# Metabolome Profiling of *Marrubium peregrinum* L. and *Marrubium friwaldskyanum* Boiss Reveals Their Potential as Sources of Plant-Based Pharmaceuticals

**DOI:** 10.3390/ijms242317035

**Published:** 2023-12-01

**Authors:** Donika Gyuzeleva, Maria Benina, Valentina Ivanova, Emil Vatov, Saleh Alseekh, Tsvetelina Mladenova, Rumen Mladenov, Krasimir Todorov, Anelia Bivolarska, Plamen Stoyanov

**Affiliations:** 1Department of Botany and Biological Education, Faculty of Biology, University of Plovdiv “Paisii Hilendarski”, 24 Tsar Assen Str., 4000 Plovdiv, Bulgaria; donikag@uni-plovdiv.bg (D.G.); cmladenova@uni-plovdiv.bg (T.M.);; 2Center of Plant Systems Biology and Biotechnology, 14 Sveti Kniaz Boris I Pokrastitel Str., 4023 Plovdiv, Bulgaria; 3Max Planck Institute for Molecular Plant Physiology, 1, Am Mühlenberg, 14476 Potsdam, Germany; 4Department of Bioorganic Chemistry, Faculty of Pharmacy, Medical University of Plovdiv, 15A Vasil Aprilov Blvd., 4002 Plovdiv, Bulgaria; 5Department of Medical Biochemistry, Faculty of Pharmacy, Medical University of Plovdiv, 15A Vasil Aprilov Blvd., 4002 Plovdiv, Bulgaria

**Keywords:** metabolome profiling, primary metabolites, secondary metabolites, flavonoids, flavonoid derivatives

## Abstract

*Marrubium* species have been used since ancient times as food additives and curative treatments. Their phytochemical composition and various pharmacological activities were the focus of a number of scientific investigations but no comprehensive metabolome profiling to identify the numerous primary and secondary metabolites has been performed so far. This study aimed to generate a comprehensive picture of the total metabolite content of two *Marrubium* species—*M. peregrinum* and *M. friwaldskyanum*—to provide detailed information about the main primary and secondary metabolites. In addition, the elemental composition was also evaluated. For this purpose, non-targeted metabolomic analyses were conducted using GC-MS, UPLC-MS/MS and ICP-MS approaches. Nearly 500 compounds and 12 elements were detected and described. The results showed a strong presence of phenolic acids, flavonoids and their glucosides, which are generally of great interest due to their various pharmacological activities. Furthermore, tissue-specific analyses for *M. friwaldskyanum* stem, leaves and flowers were carried out in order to outline the sources of potentially important bioactive molecules. The results generated from this study depict the *Marrubium* metabolome and reveal its dual scientific importance—from one side, providing information about the metabolites that is fundamental and vital for the survival of these species, and from the other side, defining the large diversity of secondary substances that are a potential source of phytotherapeutic agents.

## 1. Introduction

Metabolomics as an advanced analytical, high-throughput tool has been applied as a major approach in the comprehensive metabolite profiling and evaluation of many medicinal plants in our efforts to reveal the bases of their pharmacological activities. The huge array of phytochemical data generated by these studies is linked to the potential prediction of the biological effects according to the compound-class affiliation and respective chemical groups. Therefore, the implementation of metabolomics represents a valuable source of biomarker authentication and pharmaceutical plant evaluation [1,2]. Furthermore, it becomes an essential tool in clinical research and new drug discovery since it allows a broad and detailed assessment of the cell matrix in response to physiological changes caused by various stimuli.

Medicinal plants have been widely used in traditional medicine around the world since ancient times, leading to the recognition and utilization of many compounds of natural origin as the bioactive ingredients of modern day drugs [3]. In recent years, the development of pharmaceutical products based on natural sources has increased significantly [4,5] and, along with this, the focus on their phytochemical characterization became essential. 

The *Marrubium* genus (Lamiaceae) includes around 40 species spread across Europe, South America, the Mediterranean and Asia [6,7,8]. In the Bulgarian flora, four species have been described—*M. friwaldskyanum* Boiss., *M. vulgare* L., *M. peregrinum* L., *M. parviflorum,* Fisch et Mey [9]. Among them, *M. friwaldskyanum* is the rarest one and endemic to the country [10]. 

The literature describes the vast potential of *Marrubium* extracts and their use as herbal supplements [11]. Cytotoxic, antioxidant and antimicrobial evaluations with their respective inhibitory effects on pathological key-point regulators have been studied [12,13,14,15,16]. However, there is no comprehensive study based on an untargeted metabolomics approach that attempts to summarize the whole set of compounds available from the plant source. The literature offers a number of publications on the phytochemical composition of different *Marrubium* species [17,18,19] but very scarce information about *M. friwaldskyanum*. In the present study, we applied a metabolomics-driven non-targeted approach using several analytical platforms, including GC–MS, UPLC-MS/MS and ICP-MS, in order to give a broad description of the metabolite content of *M. peregrinum* and *M. friwaldskyanum* Boiss. plant extracts. Additionally, for *M. friwaldskyanum* Boiss, a tissue-specific comparative analysis of the leaves, stem and flowers was performed.

Primary and secondary metabolites, and some trace elements were investigated with the aim of dissecting the *Marrubium* metabolome and to eventually provide evidence about their pharmacological properties.

## 2. Results

### 2.1. GC-MS Analysis of Primary Metabolites

A total of 80 features were identified and classified into amino acids, organic acids, sugars and sugar alcohols (Appendix A). 

The principle component analysis (PCA) (Figure 1A) explained approximately 70.5% of the variation in the samples, with principle component (PC1) contributing 53.5% and principle component (PC2) contributing 17%, clearly distinguishing the four sample groups and dividing them into three distinct clusters: *M. friwaldskyanum* flowers, *M. friwaldskyanum* stem and, interestingly, *M. friwaldskyanum* leaves and *M. peregrinum*. This indicates that the primary metabolite composition of the leaves of *M. friwaldskyanum* is not significantly different from that of *M. peregrinum*.

The hierarchical cluster analysis (HCA) (Figure 1B) showed similar clustering as it recognized the *M. peregrinum* and *M. friwaldskyanum* leaves as being more similar than the species-based leaves and stem.

VIP (variable importance in projection) scores were conducted for the top 20 metabolites in order to underline the most prominent markers between all groups of samples (Figure 2) and the top 20 metabolites for each of the sample groups (Figure 3A–D) are highlighted.

The results show that the *M. friwaldskyanum* flowers were clearly distinguishable from the rest of the samples mainly due to a high accumulation of amino acids. Leucine, β-alanine, homoserine, tyrosine, acetyl-*O*-serine, valine, asparagine, isoleucine and phenylalanine were among the amino acids strongly represented in *M. friwaldskyanum* flowers and contribute as important variables in comparison with the rest of the samples.

Malic acid and its derivatives, as members of the organic acids’ class, were present, as well as some sugars and sugar alcohols like rhamnose, raffinose-like sugar, galactinol and erythritol, but the main division was at amino-acid level.

In addition, elevated amounts of ornithine, homoserine, tyrosine, arginine, glutamine, pyro glutamic acid, glycine, glycerol and isopropyl malate were detected in *M. friwaldskyanum* flowers (Figure 3A). 

The unique compound from the selected top 20, which is differentiated in terms of its discrete levels is the sugar alcohol galactinol, very often related to the protection of plant cells from oxidative damage and represented in highest concentrations in the *M. peregrinum* samples (Figure 2). 

In comparison with flowers, where the diversity of the group was mainly based on the amino acid content (nearly 80% of the highly represented markers were amino acids), the leaves’ top 20 metabolites marker distribution included changes of approximately 45% in sugars and sugar alcohols and nearly the same percentage of amino- and organic acid mixtures (Figure 3B). 

For *M. friwaldskyanum* stems, the sugar variation showed a tendency of a slight increase when compared to leaves and it reached 50% but the amino acid content was strongly decreased, giving more space to the organic acids (Figure 3C).

*M. peregrinum* formed a cluster with the *M. friwaldskyanum* leaf samples and based on the top 20 metabolites detected in *M. peregrinum*, we can confirm the presence of a similar proportion of organic acid variations and sugar content, as that found for *M. friwaldskyanum* leaves.

### 2.2. UPLC-MS/MS Analysis of Secondary Metabolites

UPLC-MS/MS is a powerful technique with very high sensitivity and selectivity; it was used in order to study the secondary metabolites and lipids present in both species.

A total set consisting of nearly 400 metabolites was assessed, including different compound classes (Appendix A). Of the 400, we were able to putatively annotate 320 compounds; 80 of them were labeled as unknown (Appendix A). 

PCA analysis clearly identified four different classes of samples which displayed their unique biochemical composition (Figure 4A). Interestingly, while the primary metabolite composition of *M. friwaldskyanum* leaves and *M. peregrinum* were identical and clustered together, in terms of their secondary metabolome, they differed very much, and this is easily seen in Figure 4B,C, where the top 20 metabolites with the highest contribution to PC1 and PC2, respectively, are given. The differentiation between the samples was mainly due to the flavonoid derivatives.

In parallel with the PCA analysis, a tissue-specific sample distribution was prepared (Figure 5) and the most prominent differences displayed separately (Figure 6). The results successfully underline the distinctions between the comprehensive biochemical networks occurring in the plant cells at tissue level, which might be related to the tissue-specific synthesis of particular secondary metabolites.

Phenylalanyl-methionine and taurochenodeoxycholic acid are two compounds found exclusively in *M. friwaldskyanum* flowers and not detected in any of the other samples. 

Several dipeptides, chlorogenic and hydroxygallic acid derivatives and some flavonoids such as luteoloside, apigenin, kaempferol and quercetin derivatives were also found to be in abundance in flowers (Figure 5A).

For the *M. friwaldskyanum* stem, no tissue-specific compounds were found. However, caffeic acid and its derivatives, salidroside, forsythoside (A, H) and its derivatives, riboflavin were in higher accumulation in comparison with the rest of the samples (Figure 5C and Figure 6C).

The overall results from the UPLC-MS/MS analysis revealed the abundance of several compounds with bioactive properties localized mainly in the green parts of the plants such as leaves and stem. A huge part of them are classified as flavonoid glycosides, which means that they might be considered as a potential source of therapeutic targets.

Metabolic profiling can generate good datasets covering a wide range of different compound classes and, at the same time, determine biochemical markers for certain biological activities. In the case of *Marrubium* samples, attention should also be given to the compounds represented in elevated numbers (Figure 7). Among them are metabolites with already proven bioactive characteristics such as flavonoid glycosides, including forsythoside derivatives, leucosceptoside A, kaempferol, apigenin, quercetin derivatives, isorhamnetin. The analysis shows their presence in all sample types but in different proportions (Figure 7A–D). In parallel with those molecules, there are others which clearly display tissue-related or species-related preferences. For example, glycylglycine—one of the simplest peptides and derivatives of hydroxygallic acid—was found to be in abundance only in *M. friwaldskyanum*.

Hydroxycinnamates are among the most widely distributed phenylpropanoids in plants. Caffeoyl derivatives as part of the cinnamic acid series have been assigned to both *Marrubium* species but are much scarcer in *M. peregrinum*. The plotted distributions (Figure 7A,D) show specific abundance in *M. friwaldskyanum* flowers in comparison with the green parts. 

Slightly different aspects are addressed by *M. peregrinum* which shows relatively high amounts of another type of phenylpropanoid glycoside, calceolarioside C in comparison with *M. friwaldskyanum* but contains smaller amounts of leucosceptoside A, a phenylethanoid glycoside, which is present mainly in *M. friwaldskyanum* tissues.

Alyssonoside was first described as a new phenylpropanoid glycoside in *M. alysson* in 1992. Our study confirmed its presence in both *Marrubium* species with prevalence in *M. friwaldskyanum* samples.

Forsythoside B, another phenylethanoid glycoside, noted for its strong anti-inflammatory properties was found to be in abundance in all tissues of *M. friwaldskyanum*. However, its derivatives were more abundant in *M. peregrinum*. The same tendency was observed for the rutin derivatives (kaemferol-3*-O*-rutinoside, quercetin-3-*O*-6-rhamnosyl glucoside, isorhamnetin rutinoside, isorhamnetin glucoside), with high accumulations in the stem, leaves and flowers of *M. friwaldskyanum* compared to *M. peregrinum*. 

The chemical profiling of the non-polar fraction revealed the lipid content of both species. Plant lipids are complex structures, important functional and structural components and their main role is related to energy storage and signaling. At the same time, they are a potential source of essential oils, widely used as pharmaceuticals.

Our study led to the identification of 175 lipid features, classified in 10 lipid classes (diacylglycerols, digalactosyl diacylglycerols, lysophospholipids, lysomonogalactosyl diacylglycerols, lysodigalactosyl diacylglycerols, monogalactosyl diacylglycerols, phosphatidylcholine, phospholipids, sphingolipids, triacylglycerides (Appendix A). In order to obtain an overview of the lipid-type distribution in each of our samples, we display a pie-chart with the respective percentages of each group in terms of the total lipid content. The results clearly show that the main lipid proportion belongs to only two lipid types, triacylglycerides (TAGs) and sphingolipids (SPs). The flowers of *M. friwaldskyanum* contained nearly 90% TAGs, followed by the leaves in which the proportion of TAGs:SPs was 74:22%. In the stem of *M. friwaldskyanum,* both classes were represented nearly in equal parts and an additional small proportion of 15% was lysophospholipids (LPLs). The rest of the lipid types constituted a minor portion in the range 1–7% (Figure 8).

Very similar behavior was observed for *M. peregrinum* samples where the ratio of TAGs:SPs was identical to that for *M. friwaldskyanum* leaves—66:26% (Figure 9).

### 2.3. Mineral Contents

The accurate assessment of the elemental composition is an important factor in the approval of novel pharmaceuticals. The dried plant samples were initially investigated for the presence of 19 elements. Of those, seven were below the limit of detection—bismuth (Bi), cadmium (Cd), chromium (Cr), cobalt (Co), lead (Pb), nickel (Ni) and thallium (Tl). The mean values for three replicates (*n* = 3) with the relative standard deviation for three independently prepared samples are given in Table 1.

## 3. Discussion

Several articles and reviews have focused on *Marrubium* representatives and reported different aspects of their pharmacological activities highlighting their potential medicinal value [20,21,22].

However, there has not been a comprehensive study based on an untargeted metabolomics approach trying to summarize the whole set of compounds available from the plant source. Here, we aimed to give a better insight into the *M. friwaldskyanum* metabolome and to dissect its specificity at tissue level. In parallel, a comparison with *M. peregrinum* outlined the unique characteristics of both species and serves as a base for a better understanding of their bioactive properties.

Plants produce thousands of low molecular weight organic substances. Based on their functions, they are classified into three different groups: primary, which are directly linked to the plant growth and development; secondary, which play a role in the plant–environmental interactions in nature, and hormones, which regulate the physiological processes and metabolism [23].

While primary metabolites are considered essential for the plant cell, the secondary metabolites display a huge diversity which might be considered in two dimensions, from one side, the plant’s biotic and abiotic interactions in nature, and from another, the tissue-specific localization of the pathways, determining the secondary metabolism within a plant [24].

The untargeted metabolomics is an important tool to explore the tissue-specific signatures of metabolic specialization and to underline the metabolites accumulated as products of tissue-specific transcriptional regulation.

The presented results emphasize the phytochemical composition of *M. friwaldskyanum* flowers, leaves and stem and display the metabolite content of *M. peregrinum*. 

The screening of all annotated secondary metabolites showed that a huge part of the Marrubium biochemical pathways are directed at the synthesis of phenolic compounds, represented by phenolic acids and flavonoids, both considered as great antioxidants. Comparative analysis revealed a much larger proportion of phenolic acids in *M. friwaldskyanum* flowers, while *M. friwaldskyanum* leaves and stem showed nearly the same distribution as in *M. peregrinum* (Appendix A). The differences between *M. friwaldskyanum* and *M. peregrinum* are generally qualitative, and seen mainly in the type of flavonoid derivatives, some of them unique for each species (secologanic acid, forsythoside D, hypoxanthine, myricetin-3-*O*-glucoside specific for *M. friwaldskyanum*, isovitexin and procyanidin B2 for *M. peregrinum*).

### 3.1. M. friwaldskyanum Flowers

The most prominent reported features from the primary and the secondary metabolite sets give the individual characteristics of each of the samples. Flower chemistry, for instance, seems to be focused on amino acids and sugar presence. This is not surprising since this part of the metabolism should sustain different physiological processes related to flower development, followed by fruit transition and seed formation [25]. Additionally, it is supposed to ensure the synthesis of specific signaling molecules which will give the specific colors and fragrances, responsible for animal pollinator interactions and finally for plant reproduction [26].

In general, a large part of the biologically active substances derived from amino acid precursors and some non-proteinogenic amino acids such as homoserine and ornithine, found in abundance in *Marrubium* flowers, might serve as intermediates for the synthesis of small bioactive peptides [27,28]. L-phenylalanine is one of the key amino acids in relation to the phenolic secondary metabolites synthesized via shikimate biosynthesis and its accumulation in flowers is not causal. The shikimate pathway is highly conserved and it represents the core unit for the phenylpropanoid metabolism responsible for the biosynthesis of a myriad of aromatic compounds such as alkaloids, flavonoids, lignins, coumarins and aromatic antibiotics [29,30].

Based on the conducted analysis, we wanted to find out the tissue-specific metabolites and to better understand the general organ-specific phytochemical composition of both plant species. 

The results clearly show that *M. friwaldskyanum* flowers are rich in amino acids and dipeptides, which most probably act as metabolic regulators [31]. Different dipeptides have been known for their health-related characteristics even though their mode of action is not well understood [32,33].

Interestingly, along with the dipeptide phenylalanyl-methionine, another substance appears as a potential flower marker. The metabolomics features lead to its annotation as taurochenodeoxycholic acid, which the literature reports as one of the main components of bile acid, with proven immunomodulatory and anti-inflammatory properties [34]. Numerous studies reveal taurochenodeoxycholic acid as an experimental drug associated with a hepatoprotective function and responsible for the activation of a phosphatidylinositol 3-kinase (PI3K)-dependent survival signaling pathway [35]. This tentative identification should be further evaluated in future and more focused studies. Both phenylalanyl-methionine and taurochenodeoxycholic acid were exclusively synthesized in flowers and not found in the rest of the samples.

A number of flavonoids and flavonoid glycosides have been reported in *M. friwaldskyanum* flowers together with a large number of amino acids and dipeptides. Luteoloside, for example, with a dominant presence in flowers, is a flavonoid that has been found in many Chinese herbs with diverse biological activities [36,37,38,39]. Apigenin coumaroylglucoside was found only in the flowers of *M. friwaldskyanum* and *M. peregrinum* samples but not detected in the stems and leaves of *M. friwaldskyanum*.

Hydrocinnamates and derivatives of hydroxygallic acid were also accumulated in the flowers. To the hydroxycinnamates, specific antioxidant activities have been assigned and a decrease in proinflammatory lysophasphatidyl choline production [40]. A very recent study has shown that hydroxygallic acid derivatives are responsible for enhanced stress resistance in *C. elegans*, and thus display neuroprotective functions [41].

Kaempferol, quercetin, isorhamnetin, apigenin derivatives, as well as flavonoid glucosides such as forsythoside, leucosceptoside, calceolarioside and their derivatives, are among the well-known bioactive substances represented in different proportions in *M. friwaldskyanum* tissue samples. In addition, *M. friwaldskyanum* seems to accumulate more rutin derivatives in comparison with *M. peregrinum* which might increase its potential bioactive properties since the oral administration of rutin leads to quercetin release, which is the main molecule responsible for the rutin properties.

### 3.2. M. friwaldskyanum Leaves

Secondary metabolites found only in *M. friwaldskyanum* leaves were identified as secologanic acid, a rare terpene molecule described in honey [42] and forsythoside D, a phenylethanoid glucoside with antioxidative and anti-inflammatory characteristics and noticeable effects of cardiovascular and neuroprotection [43]. Other substances that were present predominantly in the leaves were calceolarioside A or B, hypoxanthine, majonoside and isorhamnetin derivatives.

Isorhamnetin is a flavonoid compound found to be one of the most important ingredients in the leaves of *Ginko biloba* with proven cardiovascular and cerebrovascular protection, antitumor, anti-inflammatory and antioxidative properties [44]. In the leaves of *M. friwaldskyanum,* the mass spectral data indicated a relatively high abundance for its glucosides isorhamnetin—3-*O*-rutinoside and isorhamnetin—3-*O*-glucoside.

### 3.3. M. friwaldskyanum Stem

The highest amounts of caffeic acid were detected in *M.friwaldskyanum* stem despite the fact it did not show an individual tissue-specific metabolic fingerprint. Additionally, salidroside and alyssonoside showed prevalence in the stem in comparison with the rest of the samples. Salidroside is an bioactive substance described in *Rhodiola rosea* and used to treat Alzheimer’s disease, cancer and recently found active in cardiovascular disorders via antioxidative mechanisms [45]. Alyssonoside is another phenylethanoid glucoside with proven antioxidant and anti-inflammatory activities [46]. Notable also was the presence of riboflavin, known as vitamin B2, one of the eight essential water-soluble vitamins with crucial roles for the human body, used as treatments for and prevention of a wide array of diseases such as anemia, cancer, hyperglycemia, diabetes and many others [47].

### 3.4. M. peregrinum

Isovitexin, procyadin B and narginin dihydrochalcone are flavonoids detected exclusively in *M. peregrinum* samples. Interestingly, isovitexin was reported to be among the active components of many Chinese and medicinal plants with a wide range of pharmacological properties and recent studies suggest its use as a substitute, or adjuvant of drugs or heath-related products [48].

Procyadin B was found to be one of the most active antitumor agents isolated from natural sources [49]. Other studies report its use as a promotor of hair growth mechanisms [50].

Narginin dihydrochalcone derives from narginin, which is one of the main glycosides described in tomatoes, grapefruit and many citrus fruits, and it was shown to act as a supressor in the NF-kB signaling pathway, which is a key mediator of the inflammatory response [51,52].

In addition to these three compounds detected only in *M. peregrinum*, the presence of several others form the specific metabolic fingerprint of this species. Scopoletin and rutin derivatives, as well as the phenylethanoid glycoside forsythoside D, and the phenylpropanoid glycoside calceolarioside C, caffeolquinic and kaempferol derivatives have been detected in great abundance in *M. peregrinum* samples. This finding shows that, despite the same genus affiliation, both *M. friwaldskyanum* and *M. peregrinum* are characterized with a specific secondary metabolome.

### 3.5. Mineral Content

Micronutrients, often nominated as trace elements are esseantial for plant functions since they are directly involved in plant growth and development, affecting also plant desease control. Their most important role is related to the modulations of the biochemical reactions as they often serve as coenzymes and act as “electron carriers” in the enzyme systems [53]. The plant intake of these micronutrients depends strongly on the soil composition they grow in and represents the integrated link with human welfare when used as a food source.

As can be seen in Table 1, Ca, K and Mg were the major mineral constituents of the analyzed samples, with concentrations increasing thus Ca < Mg < K. The concentrations of the essential trace elements can be arranged as follows: Cu < Mn < Fe < Zn. The mean Zn content in the *Marrubium friwaldskyanum* flowers reached 581.3 ± 2.4 mg/kg, which is above the average Zn content for plant tissue [53]. Similar findings [54,55] indicated that species from the *Marrubium* genus have the capacity to accumulate Zn, Fe, Cu, Cd, Pb and Bi. The heavy metal content of the analyzed samples revealed that Fe was present in the highest amount in the leaf samples of *M. friwaldskyanum* with a value of 73.9 mg/kg. Cu accumulated mainly in the flower tissues of the *M. friwaldskyanum* plants (4.4 mg/kg), and small amounts were translocated to the leaves and stems. They were all within the permitted range of FAO/WHO regulatory standards [FAO/WHO] which would render them safe for further processing and use.

## 4. Materials and Methods

### 4.1. Plant Material

Plant material was collected directly from nature, in the open fields of the west region of the Rhodope mountains, during its flowering season in the period August–September of 2021 and specimens were deposited at the Herbarium of the Agricultural University (SOA), Plovdiv, Bulgaria under numbers 063315 and 063316 for *M. peregrinum* and *M. friwaldskyanum,* respectively.

*M. peregrinum* (whole plant) and *M. friwaldskyanum* (stem, leaves and flowers) were desiccated, homogenized and lyophilized in the dark, prior to further treatment and analysis.

### 4.2. Metabolite Extraction and Metabolite Measurements

Dry powder of lyophilized samples of *M. peregrinum* whole plant and *M. friwaldskyanum* leaves, stem and flowers were subjected to fractionation of metabolites during extraction, and the semipolar phase was used for GC-MS and UPLC-MS analyses, and the nonpolar one for lipid investigation (Figure 10).

Derivatization of the primary metabolites was performed according to Lisec et al. [56]. Briefly, the chemical conversion of the metabolites was achieved by treatment for 120 min at 37 °C in the presence of 40 μL 20 mg mL^−1^ methoxyamine hydrochloride (cat. no. 593-56-6, Sigma, Burlington, MA, USA) in pyridine (cat. no. 110-86-1, Merck, Rahway, NJ, USA) followed by a 30 min treatment at 37 °C with 70 μL of trimethylsilyl-N-methyl trifluoroacetamide (MSTFA, Ref. no. 701270.510, Macherey-Nagel, Düren, Germany).

A measure of 1 µL of the sample volume was injected using a Gerstel MultiPurpose system (Gerstel GmbH & Co.KG, Mülheim an der Ruhr, Germany) in splitless mode to a chromatograph coupled to a time-of-flight mass spectrometer system (Leco Pegasus HT TOF-MS; LECO Corporation, St. Joseph, MI, USA). Carrier gas was helium with a constant flow rate 2 mL s^−1^ and DB-35 column used (capillary column, 30 m length, 0.32 mm inner diameter, 0.25 μm film thickness, PN: G42, Agilent Technologies, Santa Clara, CA, USA).

The transfer line and ion source temperatures were 250 °C and the injection temperature 230 °C. The oven temperature started at 85 °C and increased at a rate of 15 °C/min^−1^ up to a final temperature of 360 °C. Mass spectra were recorded at 20 scans s^−1^ with 70–600 *m*/*z* scanning range.

The extraction was carried out according to Giavaliasco et al. [57] and Salem et al. [58] with some slight changes. Briefly, 1 mL of pre-cooled (−20 °C) methyl *tert*-butyl ether:methanol (3:1 *v*/*v*) was added to homogenized samples, which were vortexed for 1 min, incubated at 4 °C and further processed with 15 min of sonication. Phase separation was achieved by adding 500 µL of water:methanol (3:1 *v*/*v*), followed by 1 min of vortexing. Centrifuging at 11,200 rpm for 5 min was the last step. Primary and secondary metabolites, as well as lipids were analyzed using the semipolar and the organic phases, respectively. Dry polar aliquots were resuspended in water:methanol (1:1 *v*/*v*) and used for analysis of the specialized compounds. Sample component evaluation was conducted on a Thermo Scientific UPLC-ESI-Q Exactive Focus system with a HSS T3 C18 reverse-phase column (100 × 2.1 mm internal diameter, 1.8 μm particle size; Waters, Manchester, UK) that was operated at 40 °C. The mobile phase (solvent A) contained 0.1% formic acid in water and solvent B-0.1% formic acid in acetonitrile. The sample injection volume was 2 μL and the spectra were recorded in full-scan positive and negative ion modes, with a mass range from m/z 100 to 1500. The resolution used was 70,000 and the maximum scan time 250 ms. Sheath gas value was 60, and the auxiliary 35. The transfer capillary and the heater temperatures were set to 150 °C and to 300 °C, respectively. The parameters of the spray, capillary and the skimmer voltages were as follows: 3 kV, 25 V and 15 V. The MS spectra have been recorded from 0 to 19 min of the UPLC gradient. RefinerMS (version 5.3; GeneData, Basel, Switzerland) was used for chromatogram processing, peak detection and integration. Identification and further annotation of the metabolites were completed with in-house reference compound library, tandem MS (MS/MS) fragmentation, and metabolomics [59].

Analysis of the lipids was performed on the same UPLC system using a C8 reverse-phase column (100 mm × 2.1 mm × 1.7 μm particles; Waters, Manchester, UK) with dry extracts resuspended in acetonitrile:isopropanol (7:3).

The mobile phases were water (UPLC MS grade; BioSolve, Dieuze, France) with 1% 1 m NH4Ac, 0.1% acetic acid (Buffer A) and acetonitrile:isopropanol (7:3, UPLC grade; BioSolve) containing 1% 1 m NH4Ac, 0.1% acetic acid (Buffer B). A 2 μL sample volume was injected and the gradient with a flow rate of 400 μL min^−1^ was as follows: 1 min 45%A, 3 min linear gradient from 45%A to 35%A, 8 min linear gradient from 25%A to 11%A, 3 min linear gradient from 11%A to 1%A. The column was washed with 1%A for 3 min; afterwards, the buffer was set to 45%A and the column was re-equilibrated for 4 min. Total run time was 22 min. The spectra were recorded interchanging between full-scan and all-ion fragmentation-scan modes with a mass range coverage from 100 to 1500 *m/z*. Resolution used was 10,000, the capillary voltage 3 kV with sheath and auxiliary gas flow units of 60 and 35, respectively. Capillary and heater temperatures were set to 150 °C and to 350 °C; the skimmer voltage was 25 V. The spectra were recorded from minutes 1 to 17 of the UPLC gradients.

### 4.3. Compound Annotation

For the annotation of metabolites measured by GC-MS, the Golm Metabolome Database was used (Kopka et al., 2005) [60]. Lipid annotation was mainly performed by in-house library search based on full scan MS^1^, consisting of standalone standards as described in Hummel et al. (2011) [61].

### 4.4. Determination of Mineral Contents

Approximately 0.25 g of homogenized and freeze dried *M. peregrinum* and *M. friwaldskyanum* samples (flowers, leaf and stem tissues) were weighed into TFM Multiwave 3000 vessels (Anton Paar, Graz, Austria) and 0.5 mL trace metal grade concentrated HNO_3_ and 2 mL 30% H_2_O_2_ was added to each vessel. The samples were left for 30 min before being placed in the microwave and digested in closed vessels according to Miller’s [62] protocol. Duplicates of samples and method blanks were prepared and digested in a single batch and later diluted to 15 mL with reagent water.

The calibration standards and QC standards were prepared using an ICP Multi-element Standard Solution IV Certipur^®^ (Merck, Rahway, NJ, USA) and most of the elements were calibrated from 0.01 to 30 ppm. The calibration curves were based on seven standard solutions, including a blank.

The 7850 ICP-MS (Agilent Technologies, Santa Clara, CA, USA) was used for the measurement of all analytes. The system was fitted with Ultra High Matrix Introduction system and ORS4 cell operating in helium (He) mode, which reduces common polyatomic interference. The ICP-MS was fitted with glass concentric nebulizer, quartz spray chamber and torch with 2.5 mm id injector. Additionally, the operating conditions of the ICP-MS were as follows: RF power 1600 W, plasma argon flow rate 15.0 L min^−1^, nebulizer gas flow rate 0.9 L min^−1^.

### 4.5. Data Analysis

The GC and LC datasets were subjected to multivariate analysis using MetaboAnalystR [63] and “R” software, version 4.3.1.

Outliers in each variant were identified, using the 1.5 × IQR method. Individual outliers were removed, and then the values were filled with 1/5 of the minimum positive value per MetaboAnalyst’s default settings.

Lipidomics data were imported into Microsoft excel and each of the detected lipid groups was calculated as a percentage of the total lipid content for each of the samples. The data were presented as mean of six biological replicates. Primary and secondary metabolites were analyzed on both MetaboAlanyst 5.0 and R version 4.3.0 (R Core Team, 2023). PCA analysis was carried out with the help of the *prcomp(scale = TRUE)* function, from the *stats* package (R Core Team, 2023) and visualized with the *fviz_pca_ind(addElipse = TRUE, ellipse.type = “confidence”, legend.title = “Groups”, repel = TRUE*) function from the *factoextra* package [64]. Missing values were replaced with the mean value for the group they come from. Top 20 contributors to the PCA were extracted using the *fviz_contrib(choice = “var”, axes = 2, top = 20)* function from the *factoextra* package [64]. All bar charts and heat maps were prepared with the *ggplot2* package [65] and arranged with the *ggpubr* package [66].

## 5. Conclusions

The secondary metabolism of plants has a vital role for human health and life improvement. Plants remain the most important source of phytochemicals and most of them are widely used as natural phytodrugs with broad therapeutic applications. *M. friwaldskyanum* and *M. peregrinum* have rich metabolic compositions with numerous secondary metabolites, as well as being rich in the microelements essential for human health. The presence of many flavonoids such as apigenin, quercetin, rutin and their derivatives was notable, including high amounts of phenylethanoid and phenylpropanoid glycosides such as forsythoside, calceolarioside, caffeyolquinic acid, all of them with already proven antioxidant, antibacterial, anticancer and other bioactive effects. These compounds alone or in combination with others may contribute to the medicinal properties of both *Marrubium* species and may be valuable sources for new medications, and future reseach. 

In summary, *M. friwaldskyanum* and *M. peregrinum* possess:

A wide array of specialized metabolites represented mainly by phenolic acids and flavonoids, which alone predict strong antioxidant properties;

Unique, tissue-specific compounds with important bioactivities reported in the literature, which make them suitable for extraction and further application in the pharma field;

A rich source of important micronutrients such K, Mg, Ca and Zn, important for human welfare.

## Figures and Tables

**Figure 1 ijms-24-17035-f001:**
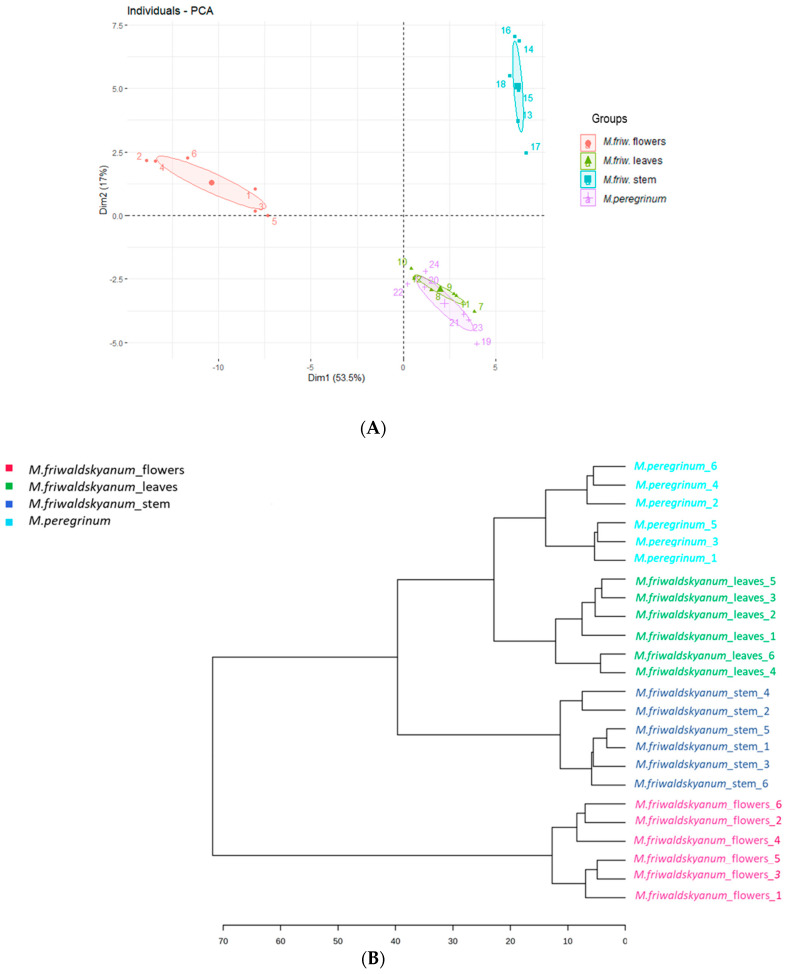
(**A**) Score plots of PCA based on the GC-MS analysis, using Ward’s clustering algorithm. The four samples are presented as follows: *M. friwaldskyanum* flowers—red circles, *M. friwaldskyanum* leaves—green triangles, *M. friwaldskyanum* stems—cyan squares and *M. peregrinum*—purple crosses. (**B**) Hierarchical cluster analysis (HCA) based on CG-MS analysis, using Ward’s clustering algorithm.

**Figure 2 ijms-24-17035-f002:**
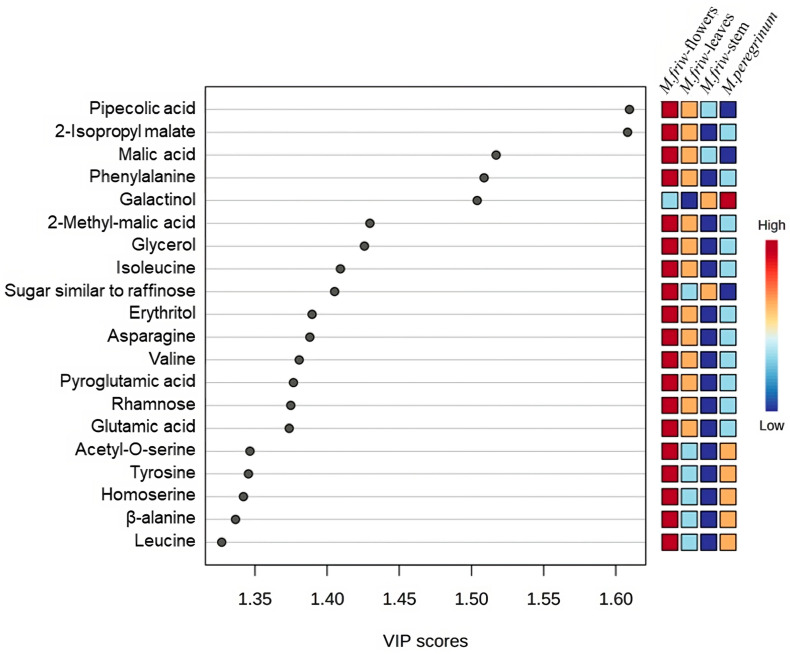
VIP scores indicate the potential markers between the four groups of samples.

**Figure 3 ijms-24-17035-f003:**
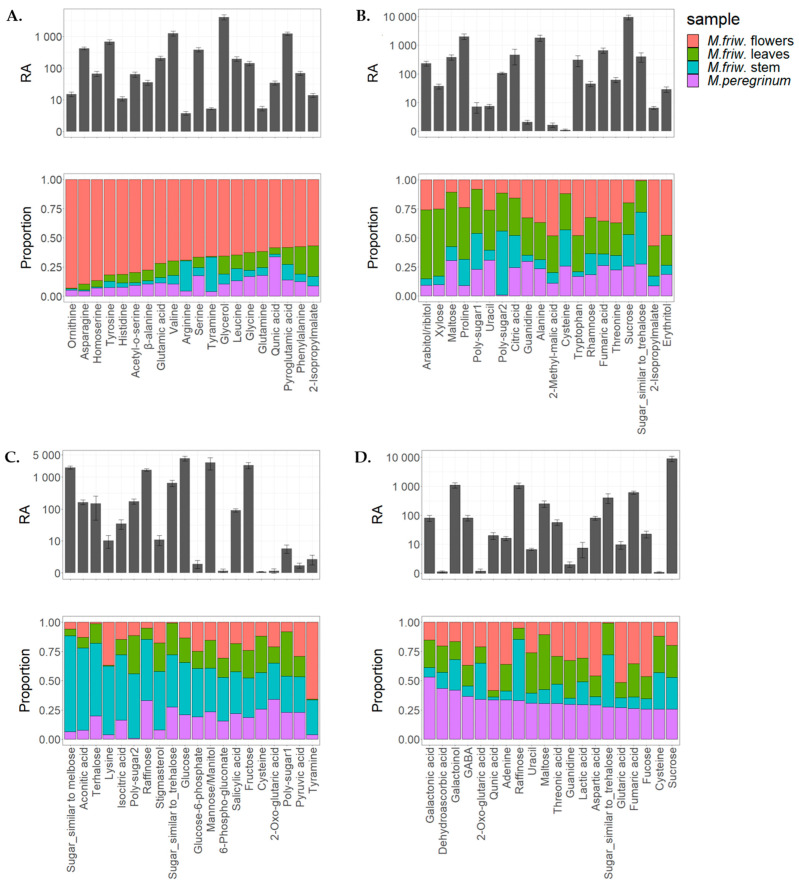
Top 20 primary metabolites’ tissue distribution in *M. friwaldskyanum* (**A**) flowers; (**B**) leaves; (**C**) stem; and (**D**) *M. peregrinum*. The four samples are presented as follows: *M. friwaldskyanum* flowers—red color, *M. friwaldskyanum* leaves—green color, *M. friwaldskyanum* stems—cyan color and *M. peregrinum*—purple color. The upper part of the figure represents the relative abundance (RA) of the top 20 primary metabolites in the respective tissues and the lower part shows their proportion, and distribution across the four sample types.

**Figure 4 ijms-24-17035-f004:**
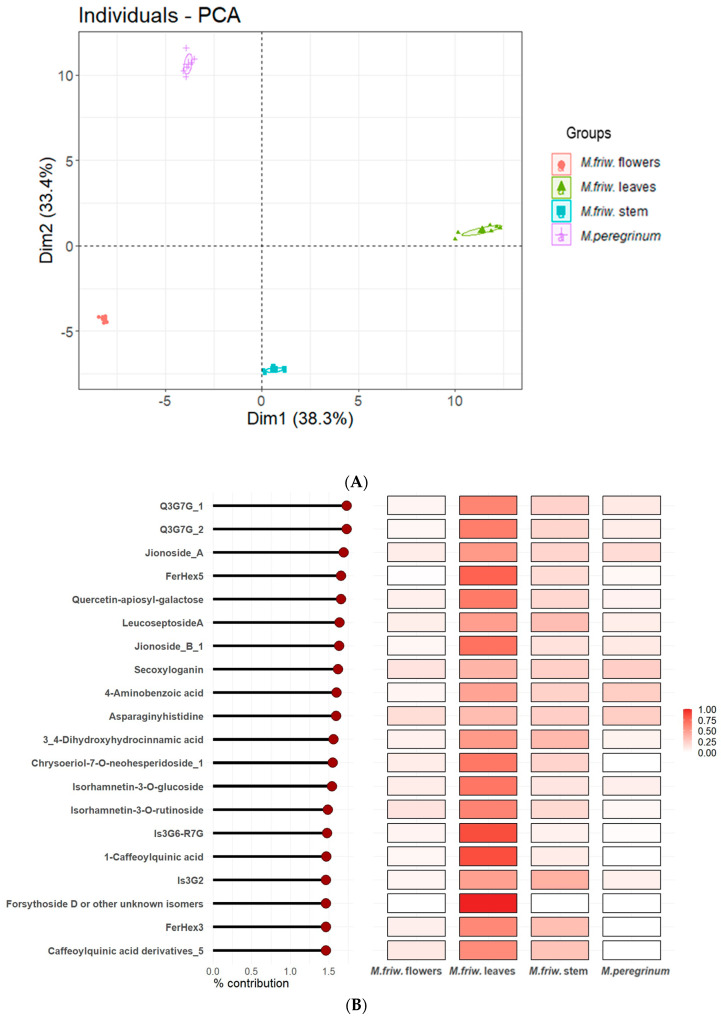
(**A**) PCA score plots based on UPLC-MS/MS analysis. The four samples are presented as follows: *M. friwaldskyanum* flowers—red circles, *M. friwaldskyanum* leaves—green triangles, *M. friwaldskyanum* stems—cyan squares and *M. peregrinum*—purple crosses. (**B**) Top 20 metabolites with highest contribution to PC1. (**C**) Top 20 metabolites with highest contribution to PC2. Abbreviations used for the compounds: Q3G7G—quercetin-3-*O*-(2″-*O*-apiosyl-6-*O*″-rhamnosyl)glucosidase; FerHex—feruloyl-hexoside; Is3G6R7G—isorhamnetin-3-*O*-(6″-*O*-rhamnosyl)glucoside; Is3G2—isorhamnetin-3-*O*-(2″-*O*-apiosyl)glucoside; KGRApCou—kaempferol-3-*O*-Glc-2″-*O*-Api-6″-*O*-Rha-pCou-7-*O*-Glc).

**Figure 5 ijms-24-17035-f005:**
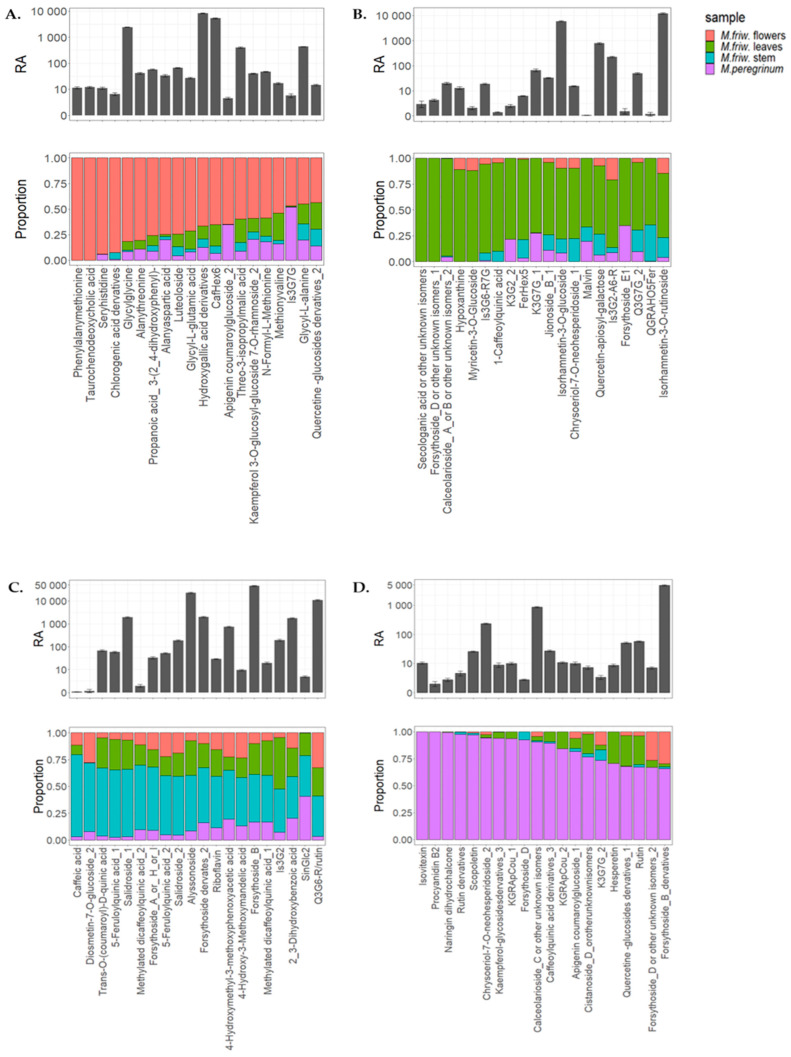
Secondary metabolites’ tissue-specific distribution in *M. friwaldskyanum*: (**A**) flowers; (**B**) leaves; (**C**) stem; (**D**) *M. peregrinum*. The four samples are presented as follows: *M. friwaldskyanum* flowers—red color, *M. friwaldskyanum* leaves—green color, *M. friwaldskyanum* stems—cyan color and *M. peregrinum*—purple color. The upper part of the figure represents the relative abundance (RA) of the top 20 secondary metabolites in the respective tissues and the lower part shows their distribution across the four sample types, where 1 is the sum of signals from the four samples.

**Figure 6 ijms-24-17035-f006:**
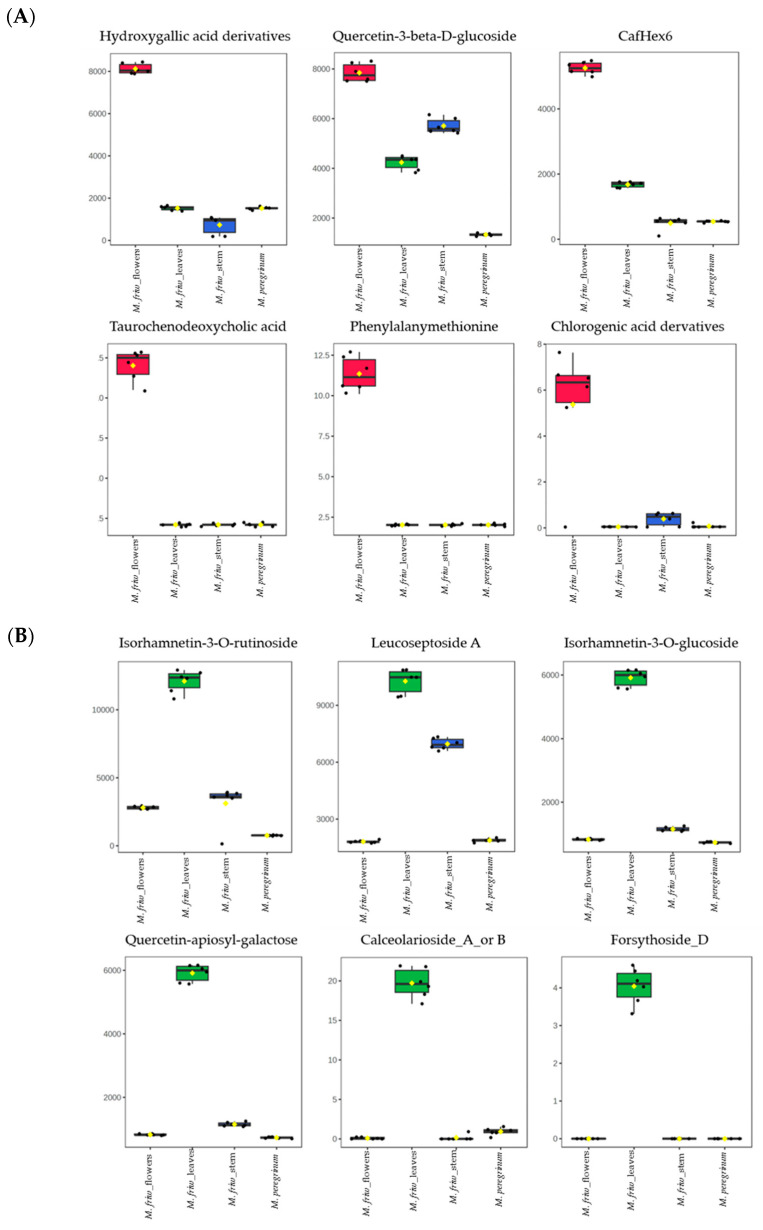
The most prominent metabolites in *M. friwaldskyanum* (**A**) flowers; (**B**) leaves; (**C**) stem; in (**D**) *M. peregrinum.* The y-axes represent the range of the relative abundances for each of the selected metabolites across all sample types.

**Figure 7 ijms-24-17035-f007:**
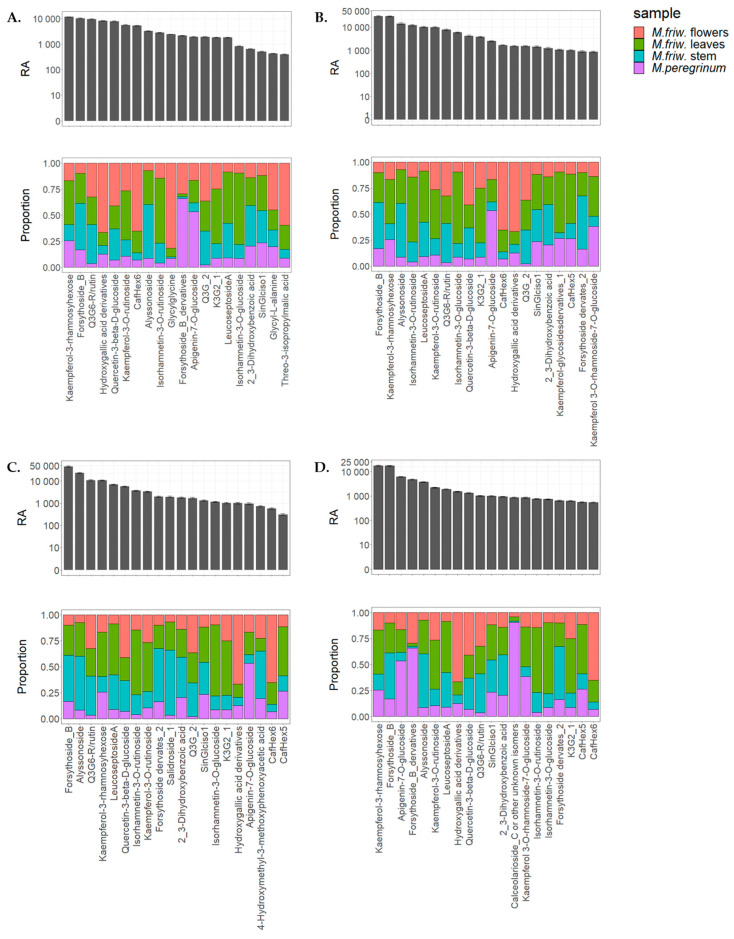
Tissue/sample distribution of highly represented secondary metabolites in *M. friwaldskyanum* (**A**) flowers; (**B**) leaves; (**C**) stem; and (**D**) *M. peregrinum.* The four samples are presented as follows: *M. friwaldskyanum* flowers—red color, *M. friwaldskyanum* leaves—green color, *M. friwaldskyanum* stems—cyan color and *M. peregrinum*—purple color. The upper part of the figure represents the relative abundance (RA) of the top 20 most abundant secondary metabolites in the respective tissues and the lower part shows their distribution across the four sample types, where 1 is the sum of signals from the four samples.

**Figure 8 ijms-24-17035-f008:**
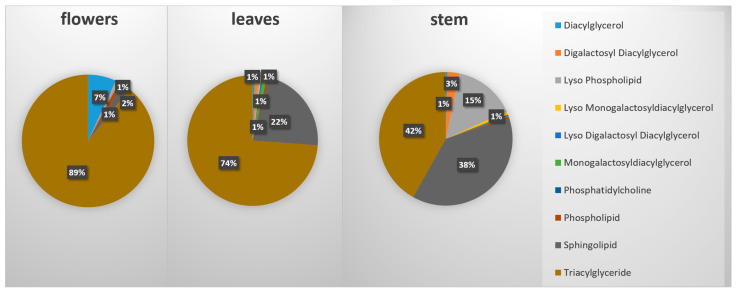
Lipid content in *M. friwaldskyanum* tissue samples.

**Figure 9 ijms-24-17035-f009:**
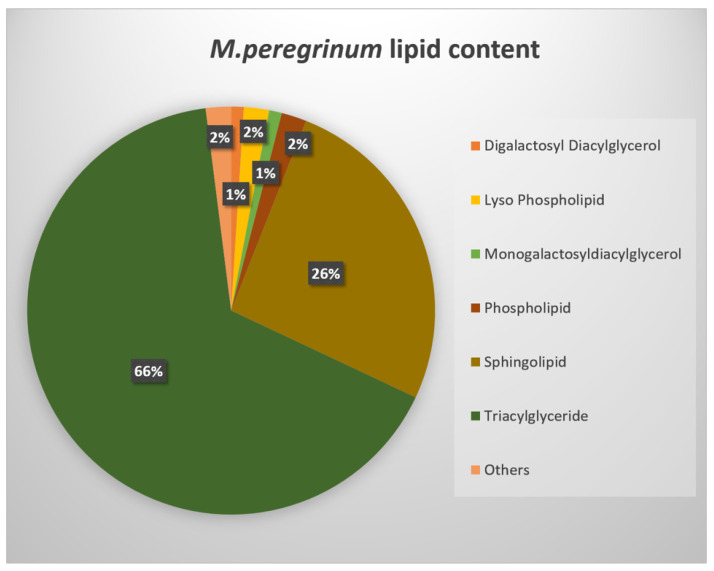
Lipid content in *M. peregrinum.* The group name “Others” includes the following lipid classes, reppresented less than 1%: Diacylglycerols, Lyso Monogalactosyldiacylglycerols; Lyso Digalactosyldiacylglycerols; Phosphatidylcholine.

**Figure 10 ijms-24-17035-f010:**
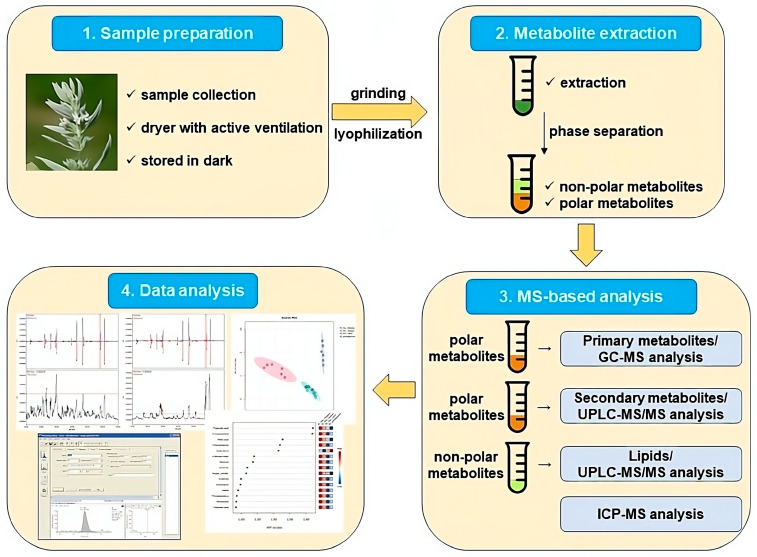
A schematic diagram of the experimental pipeline, including the steps of sample preparation, differential extraction, MS-based and data analysis.

**Table 1 ijms-24-17035-t001:** Concentrations (in mg/kg) of elements in analyzed *Marrubium* samples. Average values (*n* = 3) with relative standard deviation (RSD) in brackets.

	Ca	K	Mg	Na	B	Al
*Marrubium friwaldskyanum* flowers	187.7 (1.8)	9974.6 (1.9)	1726.3 (1.1)	12.4 (3.4)	3.9 (1.2)	12.6 (0.3)
*Marrubium friwaldskyanum* leaves	304.1 (1.2)	8212.9 (0.8)	2576.7 (0.8)	8.9 (1.1)	3.1 (0.6)	27.4 (0.1)
*Marrubium friwaldskyanum* stems	98.1 (1.0)	8945.9 (0.6)	824.3 (0.9)	6.9 (2.3)	4.1 (2.5)	4.7 (0.8)
*Marrubium peregrinum*	247.6 (1.5)	8787.8 (1.2)	1548.6 (2.1)	14.1 (2.4)	7.5 (3.7)	29.0 (1.0)
	**Mn**	**Fe**	**Cu**	**Zn**	**Sr**	**Ba**
*Marrubium friwaldskyanum* flowers	16.3 (0.8)	32.1 (0.7)	4.4 (2.2)	581.3 (2.4)	9.0 (1.8)	11.1 (0.7)
*Marrubium friwaldskyanum* leaves	26.3 (2.4)	73.9 (1.8)	2.7 (0.8)	559.8 (1.0)	9.4 (1.8)	10.4 (0.5)
*Marrubium friwaldskyanum* stems	9.1 (2.7)	9.4 (0.3)	2.3 (1.5)	472.9 (1.5)	7.8 (3.8)	14.3 (2.5)
*Marrubium peregrinum*	9.7 (1.5)	57.3 (1.8)	2.0 (1.7)	475.7 (3.7)	3.6 (3.4)	2.2 (1.1)

K, Mg, Zn and Ca have been found to be the most abundant metal constituents. *M. friwaldskyanum* leaves were found to be the richest in Ca and Mg. The highest value of K was detected in *M.friwaldskyanum* flowers. Zn was almost equally represented in M. friwaldskyanum flowers and leaves. The lowest level of detection for all micronutrients was for boron (B).

## Data Availability

Data is contained within the article or Appendix A.

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
