# Peer review of "Metabolome Profiling of Marrubium peregrinum L. and Marrubium friwaldskyanum Boiss Reveals Their Potential as Sources of Plant-Based Pharmaceuticals"

_ijms, 2023, doi:10.3390/ijms242317035_

Round 1

Reviewer 1 Report

Comments and Suggestions for Authors

Please see attached a few recommendations from the reviewer to the authors

Comments on the Quality of English Language

No comments

Reviewer 2 Report

Comments and Suggestions for Authors

The manuscript contains valuable data on the phytochemical composition of selected Marrubium species; however, it is poor described and sometimes misleading. First of all, the general purpose of the work is unclear. Why did the authors compare the composition of leaves, stems, and flowers of one species with the whole plant of another species? In my opinion, this approach is incorrect. Alternatively, please provide a detailed explanation for the purpose of such a comparison.

The manuscript can be considered  after a thorough revision. Below is a list of the other comments:

Abstract:

1)      „main primary and secondary metabolites, including elemental composition” – elemental composition is not typical primary (or secondary) metabolites. Please reedit the sentence.

2)      add some more information about the results

Keywords: remove „(List three to ten pertinent keywords….”

Introduction:

1)      „The literature describes the vast potential of Marrubium extracts and their use as herbal supplements [4-5].” – this sentece should be moved to the further part Introduction which is devoted the characteristic of Marrubium.

2)      „a number of biological activities” – specify what activities?

3)      Line 57-60:Does this part concern Marrubium? lack of references

4)      The selection of M. peregrinum and M. friwaldskyanum for the assay should be more justified. Why did Authors focused on these species?

5)      Line 66: „Primary and secondary metabolites, lipids and….” – lipids belong to primary metabolites.

6)      The aim of the study should be described more precise. In my opinion the expression „deeper knowledge about the plant metabolic biochemical processes” is exaggerated. No processes were investigated.

2. Results

Result section should be reedited. It contains many unnecessary information (or speculation) - they should be rather a part of discussion (e.g. lines 81-86;  94-97, 171-175, 294-297….). Moreover:

1)      „the idea to eventually trace its impact on their pharmacological properties.” – exaggerated expression.

2)      Figure 1. should be placed in Materials and Methods

3)      Line 93: “the leaves of M. friwaldskyanum are not significantly different from the M. peregrinum primary metabolite composition - revise as follows: the primary metabolite composition of  the leaves of M. friwaldskyanum is not significantly different from that of M. peregrinum

4)      Correct the numbering of the Figures.

5)      Figure 1A (in the current version, it should be 2A) – Please add an explanation of the abbreviations used in the figure legend. Figure 2B is a part of Figure 2, so a separate figure legend is unnecessary. The fonts on the figure are hardly visible. The same comments apply to the other Figures.

6)      Line 173: „in relation to their physiological status.” – what did the Authors mean? Different development state? For comparative tests, plants in the same growth phase should be collected.

7)      „In the  same time this information can be evaluated as a potential source of nutrients..” – information can be a source of nutrients?

8)      Figure 4,6 and 8 - the idea of presenting the results in the lower panels (proportion) is completely unclear. If the authors  aimed to show the distribution of the metabolites in different plant tissues, why did the authors include the results from M. peregrinum here.

9)      What was taken as 1 value? the sum of sygnals from all four samples?

10)   Figure 5. – name of the components in B and C panels are invisible

11)   Figure 7. Is the results of quantitative analysis? Lack of units on y-axis. Fonts are hardly visible

12)   Line 237: „highly detected peaks which in theory refer to the compounds represented in elevated amounts, in contrast with the less abundant” – avoid such statement. High of peak strongly depends on detection type. Some components are poorly ionizable and are not detected in MS

13)   Line 177: “UPLC-MS/MS” or UPLC-MS?

14)   What criteria were used for selecting the elements for elemental analysis? In my opinion, this part was unnecessarily included in the study. Moreover, Table 1 lacks statistical analysis.

Discussion

It should be rewritten. It should focused on comparison phytochemical analysis with the literature data.

1)      Lines 306-317 are unnecessary here. They could be a part of the Introduction.

2)      Lines 356-359 are unnecessary. These aspects are not linked to the present study.

3)      The section regarding the importance of elements for humans is entirely unnecessary. These aspects are not connected to the present study. If the authors intended to highlight that the plant is a rich source of the mentioned elements, they should have compared the quantities with other species.

Materials and Methods: Line 488: „C18 column” – add more detail, „gradual changes of eluent” – add gradient program (the same comments for lipid analysis). The way of identification for LC-MS is missing

Conclusion: „Plants rimain..”?

Reviewer 3 Report

Comments and Suggestions for Authors

The manuscript investigates the metabolome profiling of two species of Marrubium using GC-MS, UPLC-MS, and ICP-MS. The topic is interesting. However, I have a few observations:

1.       I recommend rereading the text for typos and grammatical mistakes.

2.       Add more details about the major findings of the study in the abstract.

3.       Remove lines 32-33.

4.       The introduction lacks a structure and doesn’t have comprehensive references to the current state of the problem. Consider rewriting.

5.       Lines 46-51 seem very general and do not serve to introduce the topic. Consider omitting this part.

6.       The figures are of low quality, and some are unreadable.

7.       Lines 84, 247, 263, and 296: avoid using references in the results section. Rewrite these parts.

8.       The discussion is very lengthy. Consolidate the arguments with a focus on the findings of this study.

9.       Materials and methods: the references appear in two different styles (e.g. Giavalisco et al. (2011)[67]). Please use the MDPI citation style.

10.   Rewrite and consolidate the conclusions in a more concise way. Focus on the major findings.

Comments on the Quality of English Language

I recommend rereading the text for typos and grammatical mistakes.

Round 2

Reviewer 2 Report

Comments and Suggestions for Authors

The authors have provided a detailed explanation and have incorporated the majority of my suggestions. However, I still have some minor comments:

Check the Latin names of the plants (especially in added fragments) – they should be italicized.

Figure 3. C, D. should be numbered as Figure 4 A, B). Alternatively, Figure c i d should constitute one panel with Figure 3 A, B (the same applies to Figure 5. C, D. and Figure 6 C, D...).

In my opinion, the concept of presenting the content of metabolites as a percentage share in the summed samples is  misleading, but I leave this point for the editor's decision.

Reviewer 3 Report

Comments and Suggestions for Authors

The authors have made substantial improvements in the revised manuscript. I think it is ready for publication

Author Response

Dear Sir/Madam,

Thank you very much for your valuable contribution during the preparation of the current MS. We believe your comments have improved its quality and we hope our research will be of interest to the broad readership of IJMS journal. 

Kind regards,

Maria Benina